# *V-PETL Bench*: A Unified Visual Parameter-Efficient Transfer Learning Benchmark

**Yi Xin**[1*], **Siqi Luo**[2,1*], **Xuyang Liu**[3*], **Yuntao Du**[4*], **Haodi Zhou**[1], **Xinyu Cheng**[1],
**Christina Lee**[5,6], **Junlong Du**[7], **Haozhe Wang**[8], **Mingcai Chen**[1], **Ting Liu**[9], **Guimin Hu**[10],
**Zhongwei Wan**[11], **Rongchao Zhang**[12], **Aoxue Li**[13], **Mingyang Yi**[14], **Xiaohong Liu**[2†]

[1]State Key Laboratory for Novel Software Technology, Nanjing University,
[2]Shanghai Jiao Tong University, [3]Sichuan University, [4]BIGAI, [5]MIT, [6]Cerebras,
[7]Tencent, [8]Hong Kong University of Science and Technology, [9]NUDT, [10]University of Copenhagen,
[11]Ohio State University, [12]Peking University, [13]Huawei Noah's Ark Lab, [14]Renmin University of China

**Project Page**: https://v-petl-bench.github.io/

## Abstract

Parameter-efficient transfer learning (PETL) methods show promise in adapting a
pre-trained model to various downstream tasks while training only a few parame-
ters. In the computer vision (CV) domain, numerous PETL algorithms have been
proposed, but their direct employment or comparison remains inconvenient. To
address this challenge, we construct a Unified Visual PETL Benchmark (V-PETL
Bench) for the CV domain by selecting **30 diverse, challenging, and compre-
hensive datasets** from image recognition, video action recognition, and dense
prediction tasks. On these datasets, we systematically evaluate **25 dominant PETL
algorithms** and open-source a modular and extensible codebase for fair evaluation
of these algorithms. V-PETL Bench runs on NVIDIA A800 GPUs and requires
approximately 310 GPU days. We release all the benchmark, making it more effi-
cient and friendly to researchers. Additionally, V-PETL Bench will be continuously
updated for new PETL algorithms and CV tasks.

## 1   Introduction

Large scale vision transformers (ViT) have achieved remarkable success in various computer vision
(CV) tasks such as image classification [1, 2, 3, 4], segmentation [5, 6, 7] and object detection [8, 9].
However, training these ViT models directly requires massive computational resources to achieve
superior performance, which is often unavailable to many academics and institutions. To alleviate
this dilemma, the "Pre-train & Finetuning" paradigm is proposed. Specifically, teams with sufficient
computational resources utilize enormous datasets [10, 11, 12, 13, 14] to train superior ViT models
and release the pre-trained weights. Researchers with limited computational resources can then
transfer the knowledge from these pre-trained ViT models to downstream tasks through a fine-tuning
stage. However, the standard Full fine-tuning, though effective, still requires substantial computational
and memory resources. This becomes particularly costly for models with billions or even trillions of
parameters. Additionally, for each task, maintaining the task-specific weights of the model brings a
storage burden, as the number of tasks increases.

---

[*]Equal contribution: xinyi@smail.nju.edu.cn, siqiluo647@sjtu.edu.cn.
[†]Corresponding author: xiaohongliu@sjtu.edu.cn.

38th Conference on Neural Information Processing Systems (NeurIPS 2024) Track on Datasets and Benchmarks.

Table 1: The comparison between V-PETL Bench and other related benchmarks.

| Benchmark | # PETL algorithms | Tasks | # Datasets | Models | Total GPU Hours |
|-----------|-------------------|-------|------------|--------|-----------------|
| ZhiJian [18] | 5 | Image Classification | 18 | ViT | - |
| V-PETL Bench | 25 | Image Recognition
Video Action Recognition
Dense Prediction | 24
3
3 | ViT
ViT, Swin
Swin | 7458 GPU Hours
(310 GPU Days) |

To mitigate the above challenges, researchers have proposed parameter-efficient transfer learning (PETL), which seeks to achieve a better trade-off between the number of trainable parameters and performance on downstream tasks. While numerous PETL algorithms for the CV domain have been proposed , their direct employment or comparison is not common. The reasons for this can be summarized as follows. First, the hyperparameters of some algorithms (e,g., learning rate, weight decay, etc.) are not open source [15, 16, 17, 18], causing subsequent researchers to spend a lot of time searching for optimal parameters. Second, the performance of baselines is often seriously underestimated in some works, making comparisons unfair. Third, existing benchmarks are mostly constrained to plain image recognition tasks, as summarized in Table 1, preventing consistent and diverse evaluation across tasks such as video action recognition and dense prediction.

To address the aforementioned issues and facilitate PETL research in the CV domain, we propose **V-PETL Bench: a Unified Visual Parameter-Efficient Transfer Learning Benchmark**. V-PETL Bench offers a *diverse* and *challenging* benchmark across 24 image recognition datasets, 3 video action recognition datasets, and 3 dense prediction datasets. Moreover, the V-PETL Bench provides comprehensive evaluations of 25 PETL algorithms by searching for hyperparameters on various pre-trained vision transformers. Since the PETL field lacks a unified evaluation metric that comprehensively considers trainable parameters and performance, we propose the Performance-Parameter Trade-off (PPT) metric to compare different algorithms using a single metric. Additionally, V-PETL Bench offers t-SNE and attention map visualizations for better analysis of PETL algorithms.

V-PETL Bench is a very *heavy-duty* and *resource-consuming* work. For the entire V-PETL Bench, we spend about 310 GPU days on NVIDIA A800 GPUs, as illustrated in Table 1. We open-source the codebase to ensure a unified and consistent evaluation of PETL algorithms. By evaluating 25 standard PETL algorithms on 30 datasets, we obtain several interesting findings: **(1)** Existing PETL algorithms can achieve performance competitive with Full fine-tuning in most downstream tasks and perform significantly better than Full fine-tuning when the amount of data is insufficient, which indicates that it could be an effective alternative to Full fine-tuning; **(2)** Existing PETL algorithms demonstrate significant efficiency, where most algorithms only updated less than 1% of the number of the pre-trained model. Additionally, they lead to improved computation and memory efficiency while achieving better performance; **(3)** Directly applying PETL algorithms from the NLP domain to vision tasks without any specific design results in performance degradation compared to well-designed PETL algorithms tailored for the CV domain; **(4)** The data and task similarity between pre-training and downstream tasks plays a key role, with higher similarity leading to better results. Furthermore, no single PETL algorithm consistently outperforms all others across all tasks.

To sum up, we list our contributions as follows:

- We propose V-PETL Bench: a unified and challenging parameter-efficient transfer learning benchmark for CV tasks for fair and consistent evaluations. To our knowledge, we are the first to build PETL benchmark that cover image classification, video action recognition, and dense prediction tasks.

- We implement 2 traditional and 25 PETL algorithms and open-source a modular codebase along with configuration files, enabling easy reproduction of the reported results in the V-PETL Bench. Our codebase is extensible and open for continued development.

- We propose the Performance-Parameter Trade-off (PPT) metric to compare PETL algorithms, which comprehensively considers two factors: task performance and trainable parameters. Additionally, we provide an in-depth analysis of these representative algorithms.

Table 2: Details of the datasets in the V-PETL Bench.

| Application | Dataset | Description | #Classes | Train size | Val size | Test size |
|---|---|---|---|---|---|---|
| | Fine-Grained Visual Classification (FGVC) [20] | | | | | |
| | CUB-200-2011 [21] | Fine-grained bird species recognition | 200 | 5,394 | 600 | 5,794 |
| | NABirds [22] | Fine-grained bird species recognition | 555 | 21,536 | 2,393 | 24,633 |
| | Oxford Flowers [23] | Fine-grained flower species recognition | 102 | 1,020 | 1,020 | 6,149 |
| | Stanford Dogs [24] | Fine-grained dog species recognition | 120 | 10,800 | 1,200 | 8,580 |
| | Stanford Cars [25] | Fine-grained car classification | 196 | 7,329 | 815 | 8,041 |
| | Visual Task Adaptation Benchmark (VTAB) [26] | | | | | |
| Image Recognition | CIFAR-100 [27] | *Natural*-tasks that contain natural images captured using standard cameras. | 100 | 800 | 200 | 10,000 |
| | Caltech101 [28] | | 102 | | | 6,084 |
| | DTD [29] | | 47 | | | 1,880 |
| | Flowers102 [23] | | 102 | | | 6,149 |
| | Pets [30] | | 37 | | | 3,669 |
| | SVHN [31] | | 10 | | | 26,032 |
| | Sun397 [32] | | 397 | | | 21,750 |
| | Patch Camelyon [33] | *Specialized*-tasks that contain images captured via specialized equipment, such as medical and satellite imagery. | 2 | 800 | 200 | 32,768 |
| | EuroSAT [34] | | 10 | | | 5,400 |
| | Resisc45 [35] | | 45 | | | 6,300 |
| | Retinopathy [36] | | 5 | | | 42,670 |
| | Clevr/count [37] | *Structured*-tasks that require geometric comprehension like object counting. | 8 | 800 | 200 | 15,000 |
| | Clevr/distance [37] | | 6 | | | 15,000 |
| | DMLab [38] | | 6 | | | 22,735 |
| | KITTI/distance [39] | | 4 | | | 711 |
| | dSprites/location [40] | | 16 | | | 73,728 |
| | dSprites/orientation [40] | | 16 | | | 73,728 |
| | SmallNORB/azimuth [41] | | 18 | | | 12,150 |
| | SmallNORB/elevation [41] | | 9 | | | 12,150 |
| Video Recognition | Kinetics-400 [42] | Video action recognition | 400 | 240,436 | N/A | 19,787 |
| | SSv2 [43] | | 174 | 168,913 | 24,777 | 27,157 |
| | HMDB51 [44] | | 51 | 3,500 | 1,500 | 1,849 |
| Dense Prediction | MS COCO [45] | Instance segmentation | 80 | 118,000 | N/A | 5,000 |
| | ADE20K [46] | Semantic segmentation | 150 | 20,000 | N/A | 2,000 |
| | PASCAL VOC [47] | Object Detection | 21 | 16,000 | N/A | 5,000 |

## 2 Related Work

As shown in Table 1, the related benchmark is ZhiJian [18]. ZhiJian includes 5 PETL algorithms but only supports image recognition tasks and the ViT model. Additionally, ZhiJian is incomplete constructed and has not been updated for a long time. Therefore, it is of significance to build a visual PETL community that can continuously update PETL algorithms to boost the development of PETL. This need is also highlighted in the survey [19]. Furthermore, Zhijian did not open source some specific details, such as parameter configurations, training logs, and model checkpoints, etc. In contrast, V-PETL Bench will open-source all these details and regularly update with new PETL algorithms and CV tasks, making it more efficient and friendly for researchers.

In the following sections, we will first introduce the downstream CV tasks and datasets, pre-trained models, PETL algorithms, and benchmark results of V-PETL Bench. Then, we will present the codebase structure of V-PETL Bench in Section 7.

## 3 Tasks and Datasets

The V-PETL Bench includes 30 datasets from image recognition, video action recognition, and dense prediction tasks, as detailed in Table 2. Each dataset in the V-PETL Bench is under a permissive license that allows usage for research purposes. These datasets are chosen based on the following considerations: (1) The dataset represents a mainstream CV task and is broadly relevant to PETL; (2) The dataset is diverse and covers multiple domains; (3) The training process is environmentally sustainable and affordable for research labs in both industry and academia.

### 3.1 Image Recognition Task

Image recognition is the primary application for PETL. The V-PETL Bench supports 24 image recognition datasets, as shown in Table 2, which can be categorized into two types as detailed below:

Table 3: Specifications of different pre-trained backbones are supported in the V-PETL Bench.

| Pre-trained Backbone | Pre-trained Objective | Pre-trained Dataset | # params (M) | Feature dim $d$ | |
|---|---|---|---|---|---|
| ViT-B [1] | | | 85 M | 768 | |
| ViT-L [1] | Supervised | ImageNet-21k | 307 M | 1024 | checkpoint |
| ViT-H [1] | | | 630 M | 1280 | checkpoint |
| Swin-B [2] | Supervised | ImageNet-22k | 88 M | 1024 | checkpoint |
| Swin-L [2] | | | 198 M | 1536 | checkpoint |
| ViT-B(VideoMAE) [50] | Self-Supervised | Kinetics-400 | 85 M | 768 | checkpoint |
| Video Swin-B [51] | Supervised | | 88 M | 1024 | checkpoint |

**Fine-Grained Visual Classification (FGVC).** FGVC comprises 5 fine-grained visual classification datasets including CUB-200-2011 [21], NABirds [22], Oxford Flowers [23], Stanford Dogs [24] and Stanford Cars [25]. If a dataset only has train and test sets publicly available, we randomly split 90% of the training set for training and 10% for validation. This validation set is then used to select hyperparameters. More details of these datasets in the V-PETL Bench can be found in Appendix B.1.

**Visual Task Adaptation Benchmark (VTAB).** VTAB comprises 19 diverse visual classification datasets, which are organized into three domains: 1) *Natural* - datasets that contain natural images captured with standard cameras. The group includes Caltech101 [28], CIFAR100 [27], DTD [29], Flowers102 [23], Pets [30], Sun397 [32], and SVHN [31]; 2) *Specialized* - datasets that contain images captured via specialized equipment, such as medical, and satellite images. The group includes Resisc45 [35], EuroSAT [34], Patch Camelyon [33] and Diabetic Retinopathy [36] ; 3) *Structured* - datasets that require geometric comprehension such as object counting. The group includes Clevr [37], dSprites [40], SmallNORB [41], DMLab [38] and KITTI [39]. Each dataset in VTAB contains 1000 training examples. Following [20], we use the provided 800-200 split of the train set to determine hyperparameters. More information on these datasets is available in Appendix B.1.

### 3.2 Video Action Recognition Task

The detailed dataset statistics for the video action recognition datasets in the V-PETL Bench are described in Table 2. We include the widely used Kinetics-400 [42], SSv2 [43], and HMDB51 [44] datasets from the previous protocol [48, 49], which are still challenging for PETL. For the SSv2 and HMDB51 datasets, we select the optimal parameters on the validation set and test the results on the test set. More details about these datasets in the V-PETL Bench can be found in Appendix B.2.

### 3.3 Dense Prediction Task

The V-PETL Bench includes three dense prediction datasets as shown in Table 2. MS COCO [45] is a representative instance segmentation dataset with 118k training images and 5k validation images. ADE20K [46] is the most widely used semantic segmentation dataset, containing 20k training and 2k validation images. Pascal VOC [47] has 16k/5k training/validation images and is used for object detection tasks. More details of these datasets in the V-PETL Bench can be found in Appendix B.3.

## 4 Pre-trained Models

In the V-PETL Bench, we experiment with the Vision Transformer (ViT) [1] and the Swin Transformer (Swin [2]), as shown in Table 3. These architectures are commonly used in the visual PETL domain. Following most research on visual PETL, we employ different levels of ViT for image recognition tasks, all pre-trained on ImageNet-21k [11]. For video action recognition tasks, we utilize the Video Swin Transformer and ViT (from VideoMAE) as the backbone. To examine the impact of pre-training data and downstream task correlation on transfer, we use pre-trained weights on Kinetics-400 [42] (a video dataset). For object detection, we use Swin-Large combined with RetinaNet [52] for training. Additionally, we employ Swin-Large with UperNet [53] for semantic segmentation tasks and Swin-Base with Cascade Mask RCNN [54] for instance segmentation tasks. For the convenience of researchers, we provide download links for all pre-trained weights.

# 5 PETL Algorithms Implemented in the V-PETL Bench

We implement 2 traditional and 25 PETL algorithms in the codebase for V-PETL Bench, including Full fine-tuning, Frozen, Adapter [15], AdaptFormer [48], SCT [55], BitFit [17], U-Tuning [56], VPT-shallow [20], VPT-Deep [20], Prefix Tuning [57], SSF [58], LoRA [16], NOAH [59], FacT [60], RepAdapter [61], Hydra [62], LST [63], DTL [64], HST [65], GPS [66], LAST [67], SNF [68], BAPAT [57], LN TUNE [69], LoRand [70], E$^3$VA [71], and Mona [72]. The algorithms are chosen based on the following considerations: 1) According to the visual PETL survey [19], existing PETL algorithms are categorized into 7 basic categories (details in Appendix C). For each category, we select 2 to 5 algorithms for implementation; 2) The algorithm is commonly used in the visual PETL domain and has considerable influence; 3) The algorithm corresponds with the comprehensive timeline of visual PETL development. More details of these algorithms can be found in Appendix D.

# 6 Benchmark Results

## 6.1 Evaluation Metrics

In the field of PETL, evaluation of algorithms typically focuses on two main aspects: the number of trainable parameters and the performance on tasks. Algorithms that achieve better performance with fewer trainable parameters generally attract more attention. However, there are currently no strict metrics to measure the PETL algorithms. To address this, we propose the **Performance-Parameter Trade-off** (**PPT**) metric. Specifically, the PPT metric for a PETL algorithm $M$ takes into account its performance $M_t$ on a downstream task $t$, its trainable parameters $P_M$, and a normalization constant $C$. The formula for $PPT_M$ is expressed as follows:

$$PPT_M = M_t \times exp(-log_{10}(\frac{P_M}{C} + 1)). \tag{1}$$

The normalization constant $C$ is set at $10^7$ as the parameters for most PETL algorithms typically fall within this range. For a detailed explanation of the design of the PPT metric, please see Appendix E.

## 6.2 Image Recognition Results

**Benchmark Results on FGVC.** The benchmark results for 13 PETL algorithms on FGVC [20] are presented in Table 4. From these results, we observe the following insights: **(1)** Compared to Full fine-tuning, PETL algorithms demonstrate competitive performance. Notably, about half of these algorithms even outperform the Full fine-tuning paradigm. **(2)** Most PETL algorithms surpass Full fine-tuning regarding PPT, highlighting their parameter efficiency. **(3)** Among the PETL algorithms, GPS [66] and SNF [68] stand out in the PPT metric. GPS achieves high performance through gradient-guided parameter selection during fine-tuning. SNF minimizes conditional mutual information to adaptively adjust the network's shortcut connections, effectively preserving important feature information. **(4)** Some PETL algorithms, such as Adapter [72], LoRA [16], and BitFit [17], originate from the natural language processing (NLP) domain. Directly applying them to vision tasks without any specific design modifications results in performance degradation.

**Benchmark Results on VTAB.** Table 5 presents the benchmark results for 18 PETL algorithms on VTAB [20]. Our analysis yields the following insights: **(1)** Almost all PETL algorithms outperform Full fine-tuning, demonstrating that fully fine-tuning the pre-trained ViT on limited data risks overfitting and catastrophic forgetting. In contrast, fine-tuning only a few parameters helps maintain the generalizability of the pre-trained models when adapting to downstream tasks. **(2)** DTL achieves the best PPT by leveraging low-rank linear mappings and feature reuse to reduce tunable parameters while enhancing performance. **(3)** Most PETL algorithms perform well on the *Natural* and *Specialized* groups because their classification objectives align with the training goals of the pre-trained dataset, ImageNet [73]. However, the *Structured* group tasks, such as object counting and depth prediction, differ significantly from ImageNet's training objectives, resulting in a substantial domain gap. PETL algorithms with less extensive parameter tuning, such as BitFit [17] and VPT-Shallow [20], fail to adequately bridge this gap, leading to sub-optimal performance.

Table 4: Benchmark results on FGVC. We evaluate 13 PETL algorithms on five datasets with ViT-B/16 models pre-trained on ImageNet-21K. We highlight the **best** and the second results.

| Method \ Dataset | CUB-200-2011 | NABirds | Oxford Flowers | Stanford Dogs | Stanford Cars | Mean | # Params. (M) | PPT |
|---|---|---|---|---|---|---|---|---|
| *Traditional Finetuning* | | | | | | | | |
| Full fine-tuning | 87.3 | 82.7 | 98.8 | 89.4 | 84.5 | 88.54 | 85.8M | - |
| Linear probing | 85.3 | 75.9 | 97.9 | 86.2 | 51.3 | 79.32 | **0 M** | 0.79 |
| *PETL Algorithms* | | | | | | | | |
| Adapter[15] | 87.1 | 84.3 | 98.5 | 89.8 | 68.6 | 85.66 | 0.41M | 0.84 |
| AdaptFormer[48] | 88.4 | 84.7 | 99.2 | 88.2 | 81.9 | 88.48 | 0.46M | 0.87 |
| Prefix Tuning [74] | 87.5 | 82.0 | 98.0 | 74.2 | 90.2 | 86.38 | 0.36M | 0.85 |
| U-Tuning [56] | 89.2 | 85.4 | 99.2 | 84.1 | **92.1** | 90.00 | 0.36M | 0.89 |
| BitFit [17] | 87.7 | 85.2 | 99.2 | 86.5 | 81.5 | 88.02 | 0.10M | 0.88 |
| VPT-Shallow [20] | 86.7 | 78.8 | 98.4 | 90.7 | 68.7 | 84.66 | 0.25M | 0.84 |
| VPT-Deep [20] | 88.5 | 84.2 | 99.0 | 90.2 | 83.6 | 89.10 | 0.85M | 0.86 |
| SSF [58] | 89.5 | 85.7 | 99.6 | 89.6 | 89.2 | 90.72 | 0.39M | 0.89 |
| LoRA [16] | 85.6 | 79.8 | 98.9 | 87.6 | 72.0 | 84.78 | 0.77M | 0.82 |
| GPS [59] | 89.9 | 86.7 | **99.7** | **92.2** | 90.4 | **91.78** | 0.66M | **0.90** |
| HST [65] | 89.2 | 85.8 | 99.6 | 89.5 | 88.2 | 90.46 | 0.78M | 0.88 |
| LAST [67] | 88.5 | 84.4 | **99.7** | 86.0 | 88.9 | 89.50 | 0.66M | 0.87 |
| SNF [68] | **90.2** | **87.4** | **99.7** | 89.5 | 86.9 | 90.74 | 0.25M | **0.90** |

Table 5: Benchmark results on VTAB. We evaluate 18 PETL algorithms on 19 datasets with ViT-B/16 models pre-trained on ImageNet-21K. We highlight the **best** and the second results.

| Method \ Dataset | Natural | | | | | | | Specialized | | | | Structured | | | | | | | | | | |
|---|---|---|---|---|---|---|---|---|---|---|---|---|---|---|---|---|---|---|---|---|---|---|
| | CIFAR-100 | Caltech101 | DTD | Flowers102 | Pets | SVHN | Sun397 | Patch Camelyon | EuroSAT | Resisc45 | Retinopathy | Clevr/count | Clevr/distance | DMLab | KITTI/distance | dSprites/loc | dSprites/ori | SmallNORB/azi | SmallNORB/ele | Mean | # Params. (M) | PPT |
| *Traditional Finetuning* | | | | | | | | | | | | | | | | | | | | | | |
| Full fine-tuning [20] | 68.9 | 87.7 | 64.3 | 97.2 | 86.9 | 87.4 | 38.8 | 79.7 | 95.7 | 84.2 | 73.9 | 56.3 | 58.6 | 41.7 | 65.5 | 57.5 | 46.7 | 25.7 | 29.1 | 65.57 | 85.8M | - |
| Linear probing [20] | 63.4 | 85.0 | 63.2 | 97.0 | 86.3 | 36.6 | 51.0 | 78.5 | 87.5 | 68.6 | 74.0 | 34.3 | 30.6 | 33.2 | 55.4 | 12.5 | 20.0 | 9.6 | 19.2 | 52.94 | **0M** | 0.53 |
| *PETL Algorithms* | | | | | | | | | | | | | | | | | | | | | | |
| Adapter [15] | 69.2 | 90.1 | 68.0 | 98.8 | 89.9 | 82.8 | 54.3 | 84.0 | 94.9 | 81.9 | 75.5 | 80.9 | 65.3 | 48.6 | 78.3 | 74.8 | 48.5 | 29.9 | 41.6 | 71.44 | 0.16M | 0.71 |
| VPT-Shallow [20] | 77.7 | 86.9 | 62.6 | 97.5 | 87.3 | 74.5 | 51.2 | 78.2 | 92.0 | 75.6 | 72.9 | 50.5 | 58.6 | 40.5 | 67.1 | 68.7 | 36.1 | 20.2 | 34.1 | 64.85 | 0.08M | 0.65 |
| VPT-Deep [20] | 78.8 | 90.8 | 65.8 | 98.0 | 88.3 | 78.1 | 49.6 | 81.8 | 96.1 | 83.4 | 68.4 | 68.5 | 60.0 | 46.5 | 72.8 | 73.6 | 47.9 | 32.9 | 37.8 | 69.43 | 0.56M | 0.68 |
| BitFit [17] | 72.8 | 87.0 | 59.2 | 97.5 | 85.3 | 59.9 | 51.4 | 78.7 | 91.6 | 72.9 | 69.8 | 61.5 | 55.6 | 32.4 | 55.9 | 66.6 | 40.0 | 15.7 | 25.1 | 62.05 | 0.10M | 0.61 |
| LoRA [16] | 67.1 | 91.4 | 69.4 | 98.8 | 90.4 | 85.3 | 54.0 | 84.9 | 95.3 | 84.4 | 73.6 | 82.9 | **69.2** | 49.8 | 78.5 | 75.7 | 47.1 | 31.0 | 44.0 | 72.25 | 0.29M | 0.71 |
| AdaptFormer [48] | 70.8 | 91.2 | 70.5 | 99.1 | 90.9 | 86.6 | 54.8 | 83.0 | 95.8 | 84.4 | 76.3 | 81.9 | 64.3 | 49.3 | 80.3 | 76.3 | 45.7 | 31.7 | 41.1 | 72.32 | 0.16M | 0.72 |
| SSF [58] | 69.0 | 92.6 | 75.1 | 99.4 | 91.8 | 90.2 | 52.9 | 87.4 | 95.9 | 87.4 | 75.5 | 75.9 | 62.3 | 53.3 | 80.6 | 77.3 | 54.9 | 29.5 | 37.9 | 73.10 | 0.21M | 0.72 |
| NOAH [59] | 69.6 | 92.7 | 70.2 | 99.1 | 90.4 | 86.1 | 53.7 | 84.4 | 95.4 | 83.9 | 75.8 | 82.8 | 68.9 | 49.9 | 81.7 | 81.8 | 48.3 | 32.8 | 44.2 | 73.25 | 0.43M | 0.72 |
| SCT[55] | 75.3 | 91.6 | 72.2 | 99.2 | 91.1 | 91.2 | 55.0 | 85.0 | 96.1 | 86.3 | 76.2 | 81.5 | 65.1 | 51.7 | 80.2 | 75.4 | 46.2 | 33.2 | 45.7 | 73.59 | 0.11M | 0.73 |
| FacT [60] | 70.6 | 90.6 | 70.8 | 99.1 | 90.7 | 88.6 | 54.1 | 84.8 | 96.2 | 84.5 | 75.7 | 82.6 | 68.2 | 49.8 | 80.7 | 80.8 | 47.4 | 33.2 | 43.0 | 73.23 | 0.07M | 0.73 |
| RepAdapter [61] | 72.4 | 91.6 | 71.0 | 99.2 | 91.4 | 90.7 | 55.1 | 85.3 | 95.9 | 84.6 | 75.9 | 82.3 | 68.0 | 50.4 | 79.9 | 80.4 | 49.2 | **38.6** | 41.0 | 73.84 | 0.22M | 0.72 |
| Hydra [62] | 72.7 | 91.3 | 72.0 | 99.2 | 91.4 | 90.7 | 55.5 | 85.8 | 96.0 | 86.1 | 75.9 | 83.2 | 68.2 | 50.9 | 82.3 | 80.3 | 50.8 | 34.5 | 43.1 | 74.21 | 0.28M | 0.73 |
| LST [63] | 59.5 | 91.5 | 69.0 | 99.2 | 89.9 | 79.5 | 54.6 | 86.9 | 95.9 | 85.3 | 74.1 | 81.8 | 61.8 | 52.2 | 81.0 | 71.7 | 49.5 | 33.7 | 45.2 | 71.70 | 2.38M | 0.65 |
| DTL [64] | 69.6 | **94.8** | 71.3 | 99.3 | 91.3 | 83.3 | **56.2** | 87.1 | 96.2 | 84.2 | | | | 48.8 | 81.9 | **93.9** | 53.9 | 34.2 | 47.1 | 74.58 | **0.04M** | **0.75** |
| HST [65] | 76.7 | 94.1 | 74.8 | **99.6** | 91.1 | 91.2 | 52.3 | 87.1 | 96.3 | **88.6** | 76.5 | **85.4** | 63.7 | 52.9 | 81.7 | 87.2 | **56.8** | 35.8 | **52.1** | **75.99** | 0.78M | 0.74 |
| GPS [66] | 81.1 | 94.2 | 75.8 | 99.4 | 91.7 | 91.6 | 52.4 | **87.9** | 96.2 | 86.5 | 76.5 | 79.9 | 62.6 | **55.0** | 82.4 | 84.0 | 55.4 | 29.7 | 46.1 | 75.18 | 0.22M | 0.74 |
| LAST [67] | 66.7 | 93.4 | **76.1** | **99.6** | 89.8 | 86.1 | 54.3 | 86.2 | 96.3 | 86.8 | 75.4 | 81.9 | 65.9 | 49.4 | **82.6** | **87.9** | 46.7 | 32.3 | 51.5 | 74.15 | 0.66M | 0.72 |
| SNF [68] | **84.0** | 94.0 | 72.7 | 99.3 | 91.3 | 90.3 | 54.9 | 87.2 | **97.3** | 85.5 | 74.5 | 82.3 | 63.8 | 49.8 | 82.5 | 75.8 | 49.2 | 31.4 | 42.1 | 74.10 | 0.25M | 0.73 |

## 6.3 Video Action Recognition Results

Table 6 displays comparative results for 5 PETL algorithms using ViT-B from VideoMAE and Video Swin Transformer on the SSv2 [43] and HMDB51 [44] datasets. The findings are as follows: **(1)** On SSv2 [43], which has sufficient data, the ViT-B from VideoMAE outperforms others, illustrating the robustness of features learned through self-supervised learning and the enhanced generalization of the pre-trained model. Conversely, on HMDB51 [44], which has limited data and fewer categories, the supervised pre-trained Video Swin Transformer shows superior performance, indicating better adaptability and generalization in smaller datasets. **(2)** On SSv2 [43], only a few PETL algorithms outperform Full fine-tuning, suggesting that with sufficient data, full fine-tuning is less likely to overfit. Conversely, on HMDB51 [44], most PETL algorithms outperform full fine-tuning, indicating that full fine-tuning may lead to overfitting when data is scarce, whereas PETL algorithms offer

Table 6: Benchmark results on SSv2 and HMDB51. We evaluate 5 PETL algorithms with ViT-B from VideoMAE and Video Swin Transformer. The results are Top-1 accuracy.

| Method | Model | Pre-training | # Params. | SSv2 Top1 | SSv2 PPT | HMDB51 Top1 | HMDB51 PPT |
|---|---|---|---|---|---|---|---|
| *Vision Transformer (from VideoMAE)* | | | | | | | |
| Full fine-tuning | ViT-B | Kinetics 400 | 85.97 M | 53.97 % | - | 46.41 % | - |
| Frozen | ViT-B | Kinetics 400 | **0 M** | 29.23 % | 0.29 | 49.84 % | 0.50 |
| AdaptFormer [48] | ViT-B | Kinetics 400 | 1.19 M | **59.02 %** | **0.56** | 55.69 % | **0.53** |
| BAPAT [57] | ViT-B | Kinetics 400 | 2.06 M | 57.78 % | 0.53 | **57.18 %** | **0.53** |
| *Video Swin Transformer* | | | | | | | |
| Full fine-tuning | Video Swin-B | Kinetics 400 | 87.64 M | 50.99 % | - | 68.07 % | - |
| Frozen | Video Swin-B | Kinetics 400 | **0 M** | 24.13 % | 0.24 | 71.28 % | **0.71** |
| LoRA [16] | Video Swin-B | Kinetics 400 | 0.75 M | 38.34 % | 0.37 | 62.12 % | 0.60 |
| BitFit [17] | Video Swin-B | Kinetics 400 | 1.09 M | 45.94 % | **0.44** | 68.26 % | 0.65 |
| AdaptFormer [48] | Video Swin-B | Kinetics 400 | 1.56 M | 40.80 % | 0.38 | 68.66 % | 0.64 |
| Prefix-tuning [74] | Video Swin-B | Kinetics 400 | 6.37 M | 39.46 % | 0.32 | 56.13 % | 0.45 |
| BAPAT [57] | Video Swin-B | Kinetics 400 | 6.18 M | **53.36 %** | 0.43 | **71.93 %** | 0.58 |

Table 7: Benchmark results on COCO. We evaluate 9 PETL algorithms with Swin-B models pre-trained on ImageNet-22K.

| Swin-B | # Params. | Memory | COCO (Cascade Mask R-CNN) $AP_{Box}$ | PPT | $AP_{Mask}$ | PPT |
|---|---|---|---|---|---|---|
| *Traditional Finetuning* | | | | | | |
| Full fine-tuning | 86.75 M | 17061 MB | 51.9 % | - | 45.0 % | - |
| Frozen | **0.00 M** | **7137 MB** | 43.5 % | 0.44 | 38.6 % | 0.39 |
| *PETL Algorithms* | | | | | | |
| Bitfit [17] | 0.20 M | 13657 MB | 47.9 % | 0.47 | 41.9 % | **0.42** |
| LN TUNE [69] | 0.06 M | 12831 MB | 48.0 % | 0.48 | 41.4 % | 0.41 |
| Partial-1 [75] | 12.60 M | 7301 MB | 49.2 % | 0.35 | 42.8 % | 0.30 |
| Adapter [15] | 3.11 M | 12557 MB | 50.9 % | 0.45 | 43.8 % | 0.39 |
| LoRA [16] | 3.03 M | 11975 MB | 51.2 % | 0.46 | 44.3 % | 0.40 |
| AdaptFormer [48] | 3.11 M | 13186 MB | 51.4 % | 0.46 | 44.5 % | 0.40 |
| LoRand [70] | 1.20 M | 13598 MB | 51.0 % | **0.49** | 43.9 % | **0.42** |
| E$^3$VA [71] | 1.20 M | 7639 MB | 50.5 % | 0.48 | 43.8 % | **0.42** |
| Mona [72] | 4.16 M | 13996 MB | **53.4 %** | 0.46 | **46.0 %** | 0.40 |

a more effective solution. **(3)** BAPAT [57] achieves outstanding performance by integrating the strengths of Adapter [15], Prefix [74], and Prompt [20].

## 6.4 Dense Prediction Results

**Benchmark Results on COCO.** Table 7 presents the results on COCO [45] using 9 PETL algorithms with pre-trained Swin-B. Our analysis reveals that: **(1)** Full fine-tuning generally outperforms most PETL algorithms. This is because COCO [45] is a substantial dataset with sufficient data, reducing the likelihood of overfitting when fully fine-tuning. However, most PETL algorithms show competitive performance, demonstrating their parameter efficiency. **(2)** Mona [72] stands out as the only PETL algorithm to surpass full fine-tuning, showcasing the effectiveness of its multi-cognitive visual filters.

**Benchmark Results on PASCAL VOC and ADE20K.** Table 8 presents the results on Pascal VOC [47] and ADE20K [46] using 9 PETL algorithms. We can observe that: **(1)** On Pascal VOC, which features fewer data and object categories, all PETL algorithms surpass Full fine-tuning. This is because adjusting a small number of parameters in the pre-trained model helps prevent overfitting and catastrophic forgetting, thereby preserving the model's generalization ability. Conversely, on ADE20K [46], which has more data and object categories, Full fine-tuning outperforms all PETL algorithms. With more available data, fully fine-tuning the pre-trained model allows for better adaptation to the

Table 8: Benchmark results on PASCAL VOC and ADE20K. We evaluate 9 PETL algorithms with Swin-L models pre-trained on ImageNet-22K.

| Swin-L | # Params. | Memory (VOC) | Pascal VOC (RetinaNet) | | ADE20K (UPerNet) | |
|---|---|---|---|---|---|---|
| | | | $AP_{Box}$ | PPT | mIoU | PPT |
| *Traditional Finetuning* | | | | | | |
| Full fine-tuning | 198.58 M | 15679 MB | 83.5 % | - | **52.10 %** | - |
| Frozen | **0.00 M** | 3967 MB | 83.6 % | 0.84 | 46.84 % | 0.47 |
| *PETL Algorithms* | | | | | | |
| Bitfit [17] | 0.30 M | 10861 MB | 85.7 % | 0.85 | 48.37 % | **0.48** |
| LN TUNE [69] | 0.09 M | 10123 MB | 85.8 % | **0.86** | 47.98 % | **0.48** |
| Partial-1 [75] | 28.34 M | **3943 MB** | 85.4 % | 0.48 | 47.44 % | 0.27 |
| Adapter [15] | 4.66 M | 10793 MB | 87.1 % | 0.74 | 50.78 % | 0.43 |
| LoRA [16] | 4.57 M | 10127 MB | **87.5 %** | 0.74 | 50.34 % | 0.43 |
| AdaptFormer [48] | 4.66 M | 11036 MB | 87.3 % | 0.74 | 50.83 % | 0.43 |
| LoRand [70] | 1.31 M | 11572 MB | 86.8 % | 0.82 | 50.76 % | **0.48** |
| E³VA [71] | 1.79 M | 4819 MB | 86.5 % | 0.81 | 49.64 % | 0.46 |
| Mona [72] | 5.08 M | 11958 MB | 87.3 % | 0.73 | 51.36 % | 0.43 |

downstream task. Nevertheless, PETL algorithms still achieve competitive outcomes, demonstrating their parameter efficiency. (**2**) LN TUNE [69] achieves the highest performance on both Pascal VOC and ADE20K, indicating that fine-tuning only the LayerNorm parameters is effective and efficient.

## 6.5 Discussion

**Computational Cost.** Some PETL works [63, 71] also explore Memory-Efficient methods, which is closely related to gradient backpropagation. As shown in Figure 1, all PETL algorithms save varying amounts of memory compared to Full fine-tuning, with Frozen and E³VA performing particularly well. The Frozen method achieves this because its backbone parameters are frozen and do not participate in gradient backpropagation.

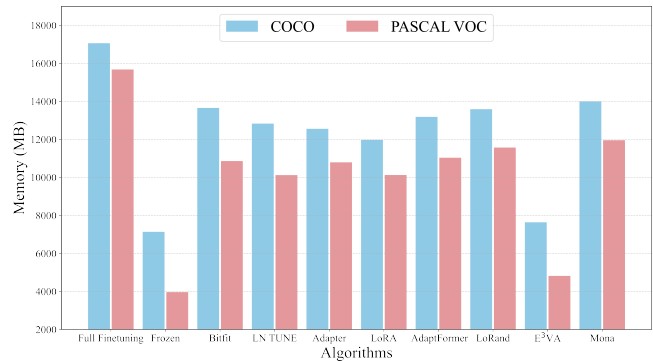

Figure 1: The GPU memory on COCO and PASCAL VOC.

E³VA designs a parallel branch for the backbone, causing the gradient backpropagation to bypass the backbone. In the future, we believe there will be more work on parameter and memory efficiency.

**Feature Distribution.** V-PETL Bench offers t-SNE visualizations that intuitively display the feature distribution for the downstream task. These visualizations enable us to evaluate the effectiveness of the PETL algorithms. Figure 2 shows t-SNE visualizations for two specific tasks, SVHN and Clevr/count, as examples. The visualizations demonstrate that the feature distribution of the data is closely linked to performance, with higher performance showing more distinct decision boundaries.

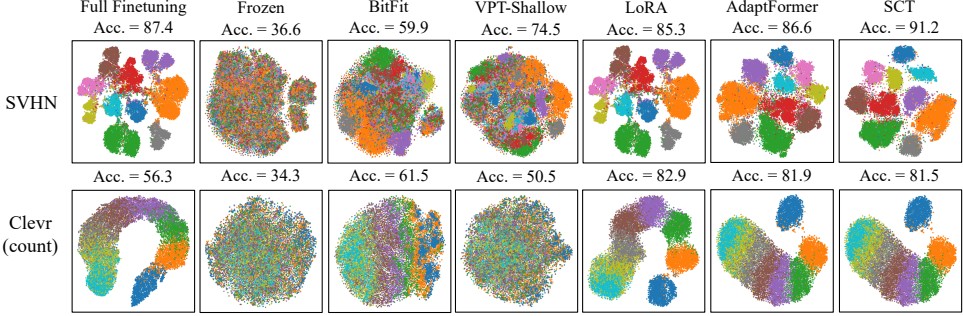

Figure 2: Visualization of feature distribution on SVHN and Clevr/count.

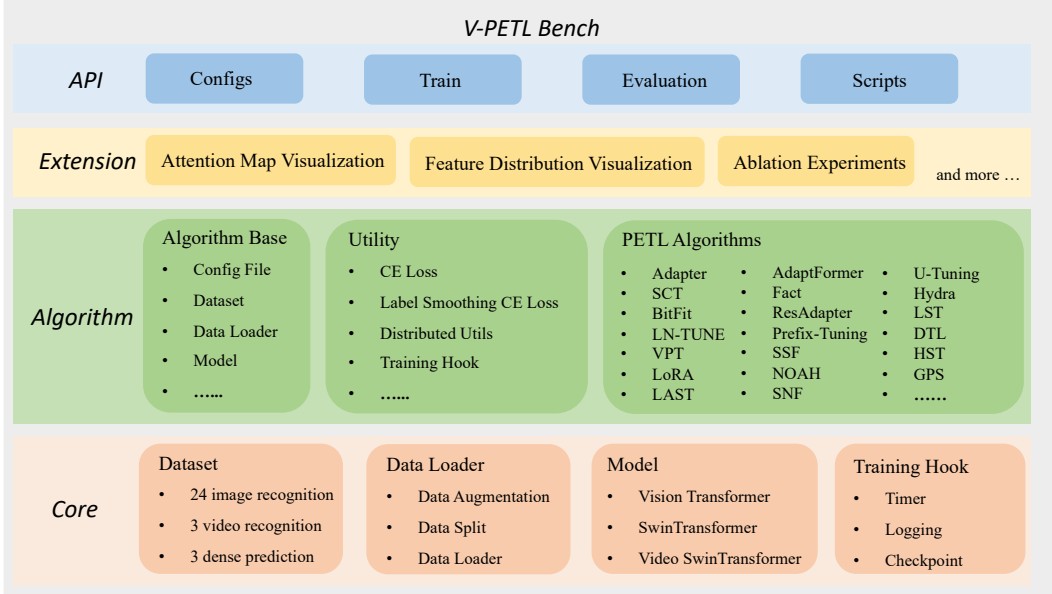

Figure 3: The structure of the V-PETL Bench codebase consists of four layers.

## 7 Codebase Structure of V-PETL Bench

In this section, we provide an overview of the codebase structure of V-PETL Bench, which which is organized into four abstract layers, as shown in Figure 3.

**Core Layer.** In the core layer, we implement the essential functions commonly used for training PETL algorithms. Additionally, this layer includes the code for datasets, data loaders, and pre-trained models that are utilized in the V-PETL Bench.

**Algorithm Layer.** In the algorithm layer, we first implement the base class for PETL algorithms, which includes initializing the datasets, data loaders, and models from the core layer. Moreover, we implement the loss functions and algorithm-specific configurations used in PETL algorithms. Based on these implementations, we currently support 25 PETL algorithms in the V-PETL Bench. More algorithms are expected to be added through the continued extension of V-PETL Bench.

**Extension Layer.** The extension layer is dedicated to advancing the core PETL algorithms for visual analysis. In this layer, we primarily implement attention map and feature distribution visualization, enabling researchers to directly observe and compare the performance of different PETL algorithms.

**API Layer.** We encapsulate the core functions and algorithms within the API layer, creating a user-friendly interface for individuals from diverse backgrounds who are interested in applying PETL algorithms to new applications. Additionally, we provide configuration files for all supported algorithms, complete with detailed parameter settings, enabling the reproduction of results.

## 8 Conclusion

In this paper, we introduce V-PETL Bench, the first comprehensive benchmark for visual parameter-efficient transfer learning domain. The V-PETL Bench includes 30 CV datasets and implements 25 dominant PETL algorithms. We also propose the PPT metric to compare different algorithms based on both the number of parameters and the performance. Additionally, we conduct several insightful analyses of the results. We regard V-PETL Bench as a long-term evolving project and are dedicated to its continuous development. Our roadmap for the future includes expanding its scope to the multimodal model and the generative model.

**Acknowledgement.** The work was supported in part by the National Natural Science Foundation of China under Grant 62301310, and in part by Sichuan Science and Technology Program under Grant2024NSFSC1426.

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
