# A  Parameter-Efficient Transfer Learning Definition

**Definition 1** (Parameter-efficient Transfer Learning). *Given a pre-trained model $M$ parametrized by $\theta$, and a specific downstream task $\mathcal{D} = \{(x_i, y_i)\}_{i=1}^{|\mathcal{D}|}$, where $(x_i, y_i)$ represents each ground-truth input-output pair related to task $\mathcal{D}$, parameter-efficient transfer learning seeks to adapt $\theta$ to task $\mathcal{D}$, where task-specific parameters increment $\Delta\theta$ is introduced with $|\Delta\theta| \ll |\theta|$. The optimal parameters are found by optimizing the losses $\mathcal{L}$ on task $\mathcal{D}$:*

$$\min_{\Delta\theta} \mathbb{E}_{(x_i,y_i)\in\mathcal{D}} \mathcal{L}(M_{\theta+\Delta\theta}(\hat{y}_i|x_i), y_i). \tag{2}$$

# B  Details of Datasets in V-PETL Bench

## B.1  Image Recognition Tasks

In this section, we introduce all the image recognition datasets in the V-PETL Bench. For each dataset, we select relevant examples, as illustrated in Figure 6. To download the datasets, please refer to project page https://v-petl-bench.github.io/.

**CUB-200-2011**  This task represents the most widely-used benchmark for fine-grained visual categorization. It includes 11,788 images across 200 subcategories of birds, with 5,994 images designated for training and 5,794 for testing.

**NABirds**  This task comprises a collection of 48,000 annotated photographs representing 400 species of birds commonly observed in North America. Each species is documented with over 100 photographs, which include separate annotations for males, females, and juveniles.

**Oxford Flowers**  The task involves an image classification dataset comprising 102 flower categories, featuring flowers commonly found in the United Kingdom. Each category contains between 40 and 258 images.

**Stanford Dogs**  The task includes 20,580 images spanning 120 classes of dogs from around the world, divided into 12,000 images for training and 8,580 for testing.

**Stanford Cars**  The task features 196 classes of cars, totaling 16,185 images captured from the rear. The categories are organized by Make, Model, and Year. Each image has a resolution of 360×240 pixels.

**Caltech101**  The task involves classifying images of objects across 101 categories, plus an additional background clutter class. The objects include a diverse range such as animals, airplanes, chairs, and scissors. Image sizes vary, typically ranging from 200 to 300 pixels per edge.

**CIFAR-100**  The task involves classifying natural images into 100 classes, with each class containing 500 training images. Examples of these classes include apples, bottles, dinosaurs, and bicycles. All images are 32x32 pixels in size.

**DTD**  The task involves classifying images of textural patterns across 47 classes, each containing 120 training images. The textures include varied patterns such as banded, bubbly, meshed, lined, and porous. Image sizes range from 300x300 to 640x640 pixels.

**Flowers102**  The task involves classifying images of flowers found in the UK into 102 categories, with each category containing between 40 and 248 training images. Examples of these flowers include Azalea, Californian Poppy, Sunflower, and Petunia. All images have a minimum dimension of 500 pixels.

**Pets** The task involves classifying images of cat and dog breeds into 37 classes, with approximately 200 images per class. Examples include Persian cats, Chihuahua dogs, English Setters, and Bengal cats. The dimensions of the images are typically 200 pixels or larger.

**Sun397** The Sun397 task is a scenery benchmark that includes 397 classes, each with at least 100 images. The classes are organized hierarchically and feature a variety of scenes such as cathedrals, staircases, shelters, rivers, and archipelagos. All images are in color and measure at least 200x200 pixels.

**SVHN** This task involves classifying images from Google's Street View of house numbers into 10 classes, each containing over 1,000 training images. Each image is 32x32 pixels in size.

**EuroSAT** The task involves classifying Sentinel-2 satellite images into 10 different land use categories, such as Residential, Industrial, River, and Highway. Each image has a spatial resolution of 10 meters per pixel and measures 64x64 pixels.

**Resisc45** The Remote Sensing Image Scene Classification (RESISC) dataset is designed for scene classification tasks using remote sensing images. It comprises 45 classes, each with 700 images, featuring diverse scenes such as tennis courts, ships, islands, lakes, parking lots, sparse residential areas, and stadiums. Each image is in RGB format with dimensions of 256x256 pixels.

**Patch Camelyon** The Patch Camelyon dataset includes 327,680 images from histopathologic scans of lymph node sections. The classification task involves predicting the presence of metastatic tissue within these images, dividing them into two classes. Each image has dimensions of 96x96 pixels.

**Retinopathy** The Diabetic Retinopathy dataset consists of image-label pairs with high-resolution retina images, and labels that indicate the presence of Diabetic Retinopahy (DR) in a 0-4 scale (No DR, Mild, Moderate, Severe, or Proliferative DR).

**Clevr/count** CLEVR is a visual question and answer dataset designed to evaluate algorithmic visual reasoning. We use just the images from this dataset, and create a synthetic task by setting the label equal to the number of objects in the images.

**Clevr/distance** Another synthetic task we create from CLEVR consists of predicting the depth of the closest object in the image from the camera. The depths are bucketed into size bins.

**dSprites/location** The dSprites dataset was originally designed to asses disentanglement properties of unsupervised learning algorithms. In particular, each image is a 2D shape where six factors are controlled: color, shape, scale, rotation, and (x,y) center coordinates. Images have 64x64 black-and-white pixels. This task consists in predicting the x (horizontal) coordinate of the object. The locations are bucketed into 16 bins.

**dSprites/orientation** We create another task from dSprites consists in predicting the orientation of each object, bucketed into 16 bins.

**SmallNORB/azimuth** The Small NORB dataset features images of 3D toys from 50 classes, including animals, human figures, airplanes, trucks, and cars, each captured in a resolution of 640x480 pixels. In this specific task, we assign labels based on the azimuth—the angle of horizontal deviation—using intervals of 20 degrees, resulting in 18 distinct classes.

**SmallNORB/elevation** Another synthetic task from the Small NORB dataset involves predicting the elevation depicted in the images. This task is divided into 9 classes, each corresponding to a different elevation ranging from 30 to 70 degrees, with intervals of 5 degrees between each class.

**DMLab** The DMLab (DeepMind Lab) is a set of control environments focused on 3D navigation and puzzle-solving tasks. The Dmlab dataset contains frames observed by the agent acting in the DeepMind Lab environment, which are annotated by the distance between the agent and various objects present in the environment. The goal is to evaluate the ability of a visual model to reason about distances from the visual input in 3D environments. The Dmlab dataset consists of 360x480 color images in 6 classes.The classes are {close, far, very far} × {positive reward, negative reward} respectively.

**KITTI-Dist** The KITTI task involves predicting the binned depth to vehicles, such as cars, vans, or trucks, in images. The prediction is categorized into four distinct bins or classes.

## B.2 Video Action Recognition Tasks

**SSv2** The SSv2 dataset currently contains 168,913 videos, which are categorized under 174 different labels. These videos vary in length from 2 to 6 seconds. The labels are textual descriptions created from templates like "Dropping [something] into [something]," where the "[something]" acts as placeholders for various objects.

**HMDB51** The HMDB51 dataset comprises a diverse array of realistic videos sourced from movies and web videos. It contains 6,766 video clips, spread across 51 action categories such as "jump," "kiss," and "laugh." Each category includes at least 101 clips, providing a broad spectrum of human actions.

## B.3 Dense Prediction Tasks

**MS COCO** The MS COCO dataset is a widely used benchmark for instance segmentation, featuring 80 object categories. Each image in MS COCO is accompanied by high-quality annotations. For instance segmentation, these annotations consist of pixel-wise masks for each object instance, enabling precise identification and localization of objects within the images.

**ADE20K** The ADE20K semantic segmentation dataset includes over 20,000 scene-centric images, each exhaustively annotated with pixel-level labels for objects and object parts. It encompasses a total of 150 semantic categories, which range from expansive elements like sky, road, and grass, to discrete objects such as person, car, and bed.

**PASCAL VOC** The PASCAL VOC dataset features 20 object categories, encompassing vehicles, household items, animals, and more. Each image in this dataset is equipped with bounding box annotations and object class annotations, making it a widely used benchmark for object detection.

# C  Taxonomy of PETL Algorithms

In visual PETL survey [19], existing PETL methods can be divided into 7 basic categories, including:

**Adapter Tuning** methods inject small-scale neural modules (adapters) to the Transformer layers and only tune these adapters for model adaptation, as showin in Figure 4b. Specifically, one adapter module contains a down-projection and an up-projection. Additionally, there is a nonlinear layer between the two layers for non-linear projection.

**Prompt Tuning** methods wrap the original input with additional visual prompts. These prompts consist of trainable parameters or perturbations, as shown in Figure 4c. Given an input $x_0 \in R^d$ and prompts $P = [P_1], [P_2], ...[P_l] \in \mathbb{R}^{l \times d}$, the final input can be expressed as follows:

$$x_0 = concat(P, x_0) = [P, x_0] \in \mathbb{R}^{(l+N) \times d}. \tag{3}$$

**Side Tuning** employs a smaller and separate network that operates in parallel with the Transformer, as shown in Figure 4d. While ensuring parameter-efficient, this separation completely obviates the

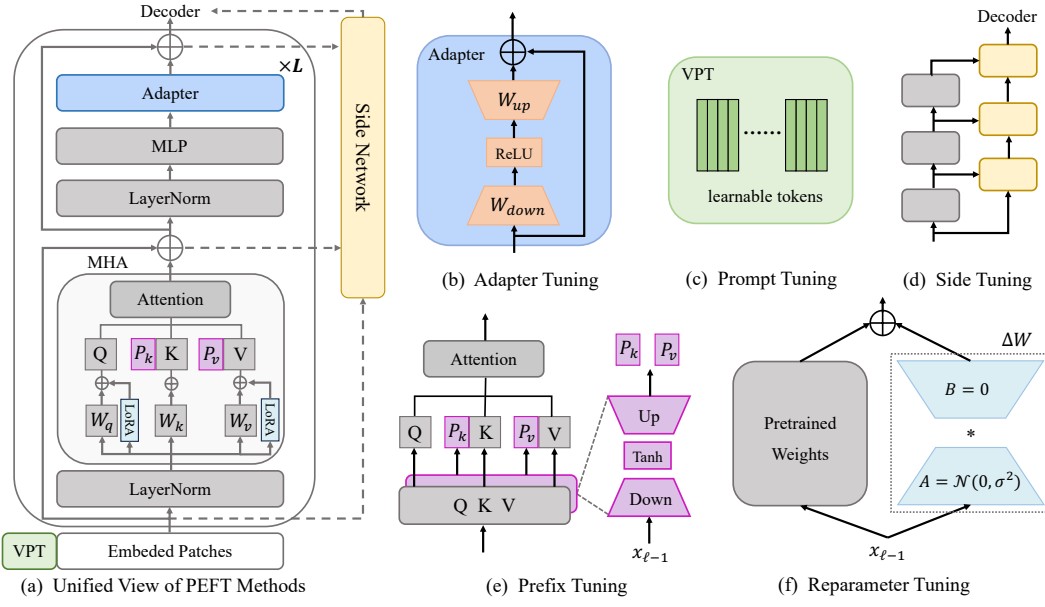

(a) Unified View of PEFT Methods      (b) Adapter Tuning      (c) Prompt Tuning      (d) Side Tuning      (e) Prefix Tuning      (f) Reparameter Tuning

Figure 4: The detailed architecture of various PEFT methods. [19]

need for costly backpropagation through a large backbone network, resulting in significant GPU memory savings.

**Prefix Tuning** introduces learnable prefix matrices to the Multi-Head Attention (MHA) module of the Transformer layers, as shown in Figure 4e. It involves prepending two randomly initialized prefix matrices $P_k, P_v \in R^{l \times d}$ to the keys and values in the MHA, leading the attention calculation to:

$$Attention(Q, K, V) = softmax(\frac{Q[P_k, K]^T}{\sqrt{d}})[P_v, V]. \tag{4}$$

**Specification Tuning** is an efficient approach that directly modifies a specific subset of parameters in Transformers, such as bias and LayerNorm, which are crucial for downstream tasks. This method concentrate on important parameters while discarding those deemed less relevant. The concept, while straightforward, has proven to be surprisingly effective.

**Reparameter Tuning** methods introduce new learnable parameters during the training stage, while these parameters can be integrated into the original Transformer layers through reparameterization during the inference phase, as shown in Figure 4f.

**Unified-based Tuning** methods offer a unified framework to integrate various fine-tuning methods into a single, harmonized architecture. This approach streamlines the process and enhances the overall efficiency and effectiveness of the fine-tuning.

## D    Details of Implemented PETL Algorithms in V-PETL Bench

For each category of PETL algorithms outlined in Section C, we select multiple algorithms for implementation within the V-PETL Bench. These are briefly introduced in Section 5 of the paper, with a detailed introduction provided below:

### D.1    Adapter Tuning

**(1) Adapter** [15] incorporates a bottleneck module (Adapter) into each Transformer layer, positioned after both the Multi-Head Attention (MHA) and the Feed-Forward Networks (FFN). **(2) Adapt-Former** [48] embeds the adapter module parallel to the FFN in each encoder of a Vision Transformer.

**(3) SNF** [68] integrates and fine-tunes Normalizing Flows modules within the residual connections of each encoder block in the pre-trained ViT. **(4) Hydra** [62] includes both a parallel and a sequential adapter in the final linear layer of the FFN in each encoder block. **(5) LoRand** [70] introduces low-rank adapters into each SwinBlock of the Swin Transformer, positioned after the MHA and FFN for enhanced dense predictions. **(6) Mona** [72] integrates a multi-cognitive convolutional filter group (Depthwise Convolution) along with an aggregation filter ($1 \times 1$ Convolution) following the down projection of the standard adapter.

## D.2 Prompt Tuning and Prefix Tuning

**(1) VPT** [20] enhances vision transformers by appending learnable visual prompts to the input sequences (VPT-Shallow) or to the input of each transformer encoder layer (VPT-Deep). **(2) Prefix Tuning** [74] incorporates trainable prefix tokens into each layer of the Transformer model, allowing for task-specific adaptation of the pre-trained model without altering the original parameters.

## D.3 Side Tuning

**(1) LST** [63] employs a small ladder-side network that operates outside the pre-trained network, receiving intermediate activations via shortcut connections. **(2) DTL** [64] integrates a Compact Side Network (CSN) alongside each encoder block within the ViT, extracting task-specific information progressively throughout the forward pass and reintegrating it into the pre-trained ViT. **(3) HST** [65] constructs a lightweight Hierarchical Side Network (HSN) separate from the pre-trained ViT to generate multi-scale features from intermediate activations. **(4) LAST** [67] adds a lightweight side network consisting of low-rank self-attention (LSA) modules after each Transformer block in the pre-trained ViT. **(5) E$^3$VA** [71] introduces a highway system parallel to the SwinBlock in the Swin Transformer, featuring trainable low-rank adapters (E$^3$VA) that isolate the pre-trained model from gradient backpropagation.

## D.4 Specification Tuning

**(1) BitFit** [17] exclusively fine-tunes the bias terms of the pre-trained model while keeping the rest of the parameters unchanged. **(2) GPS** [66] fine-tunes a small, crucial subset of parameters (sub-network) from the original pre-trained model using a gradient-based approach. **(3) SCT** [55] incorporates Salient Channel Tuning Modules (SCTM) after the MHA or FFN to target and fine-tune a select group of channels (salient channels). **(4) LN TUNE** [69] specifically fine-tunes the LayerNorm parameters, leaving other components untouched. **(5) Partial-1** [75] focuses on fine-tuning only the last encoder layer of the pre-trained ViT.

## D.5 Reparameter Tuning

**(1) LoRA** integrates a tunable pair of low-rank decomposed weight matrices into each encoder layer of the pretrained ViT. **(2) FacT** incorporates tunable factorized weight matrices into each layer of the pretrained ViT. **(3) RepAdapter** introduces a reparameterizable linear adapter before the MHA and FFN of each encoder block of the pre-trained ViT. **(4) SSF** adds a tunable scaling and shifting module (SSF-ADA) behind the MHA and FFN of the pre-trained ViT.

## D.6 Unified-based Tuning

**(1) NOAH** [59] integrates a tunable Neural Operator Adaptation Head (NOAH) module, which includes a lightweight MLP and a gating mechanism, into the MHA and FFN of each encoder block in the pre-trained ViT. **(2) U-Tuning** [56] incorporates a unified tuner (U-Tuner) consisting of Prefix, Adapter, and Prompt elements into the MHA and FFN of each encoder block in the pre-trained ViT. **(3) BAPAT** [57] introduces the Parallel Attention (PATT), which adds trainable query, key, and value matrices to the MHA, and incorporates bottleneck layers into the FFN of each Transformer block in the pre-trained ViT.

# E   Performance-Parameter Trade-off Metric

For the evaluation of PETL algorithms, to compare different methods with a single number that considers both task performance and parameter-efficiency, we define the Performance-Parameter Trade-off (PPT) metric:

$$PPT = performance \times exp(-log_{10}(\frac{\#parameters}{C} + 1)). \tag{5}$$

The performance quantifies prediction accuracy, ranging from 0 to 1. The term # parameters refers to the count of updated parameters during the model adaptation phase. C is a normalization constant set at $10^7$, aligns with the typical parameter sizes of existing PETL algorithms, ensuring that the ratio $\frac{\#parameters}{C}$ falls within the [0, 1) range. To prevent log values from reaching negative infinity, we introduce an additive constant of 1. As PETL evolves, considerations such as GPU memory might be incorporated, potentially leading to further refinements of the PPT metric.

## E.1   Attention Map

Attention map visualization is a crucial tool for analyzing PETL algorithms. In the V-PETL Bench, we have included an attention map visualization module. We randomly select several examples of attention map visualizations, as illustrated in Figure 5. We can find that the SCT method focuses more intently on the cat, while Full Finetuning directs more concentrated attention towards the flower. In the V-PETL Bench, we offer a convenient interface that allows researchers to easily utilize the attention map visualization tool.

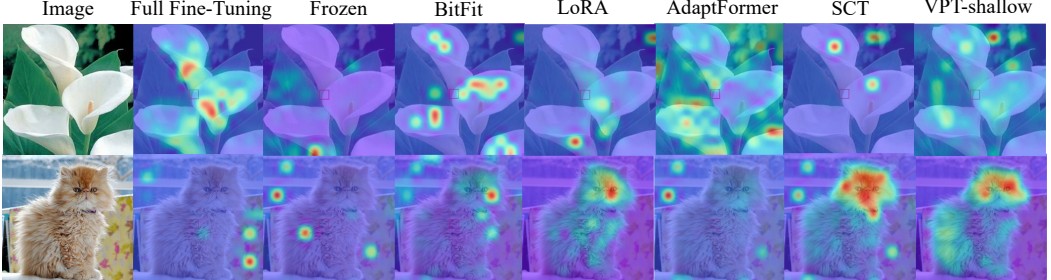

Figure 5: Visualization of attention maps.

# F   Broader Impacts

**Efficient usability.**   The V-PETL Bench provides a user-friendly calling interface and comprehensive documentation, enabling researchers from various fields to quickly get started with efficient usability. Additionally, V-PETL Bench provides checkpoints for all tasks, allowing researchers to directly load and utilize these models without the need for retraining.

**Environmental-friendly consumption.**   The V-PETL Bench has standardized common computer vision tasks and offers evaluation results for various PETL algorithms. It eliminates the need for additional training unless absolutely necessary, which positively impacts carbon emissions reduction and environmental protection.

**Ethical Considerations.**   The PETL algorithms capitalize on the representation and generalization abilities acquired from large-scale pre-trained datasets and models. However, it's crucial to acknowledge the potential risks if these pre-training datasets contain bias or illegal information.

## G    Future Work

In this work, our primary focus is on traditional computer vision (CV) tasks. However, there are additional CV tasks that should not be overlooked by the PETL community. Currently, the V-PETL Bench does not encompass tasks such as text-to-image generation, point cloud analysis, or robotic manipulation. Moreover, it supports only a limited selection of pre-trained models. Expanding our repository to include self-supervised and multimodal pre-trained models is essential. We plan to continue updating the benchmark to include these enhancements in the future.

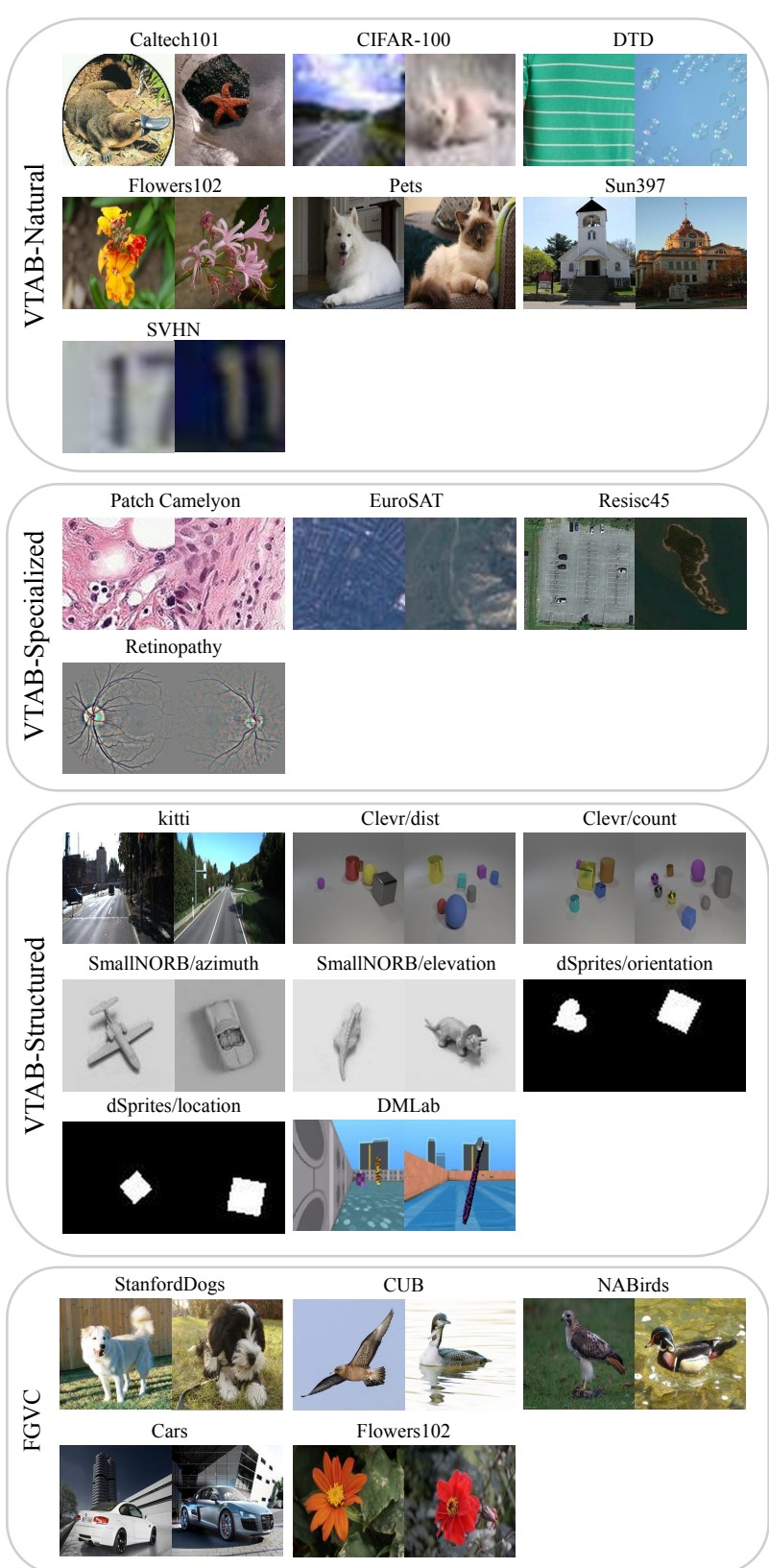

Figure 6: Dataset examples for all classification tasks evaluated.