# OpenReview forum: "V-PETL Bench: A Unified Visual Parameter-Efficient Transfer Learning Benchmark"
_NeurIPS.cc/2024/Datasets_and_Benchmarks_Track — NeurIPS 2024 Track Datasets and Benchmarks Poster_

### Official Review · Reviewer_etT1 · 2024-07-19

**Rating:** 8
**Confidence:** 3
**Correctness:** The work is open-sourced and easy to …
**Clarity:** The paper is easy to follow.

**Review:**

The benchmark introduced in this work demonstrates significant potential for tracking progress in PEFT research and facilitating fair comparisons between different algorithms. In the era of large models, PEFT techniques are crucial. I foresee this benchmark playing a pivotal role in the field of PEFT.

**Strengths:**

1. The paper is well-organized and easy to follow.
2. This work dedicates substantial GPU time to achieve the optimal performance of each PEFT algorithm, ensuring fair comparisons, which is admirable.
3. The benchmark is open-sourced and well-documented. The workload is huge.

**Additional Feedback:**

N/A

**Documentation:**

The work is well-documented in github.

**Ethics:**

There is no ethics concern.

**Limitations:**

1.It would be better to use FLOPs as a comparison metric. The number of parameters and memory do not necessarily reflect the computational effort required to fine-tune the models.

**Opportunities For Improvement:**

1. It would be better to explain the insight of the proposed evaluation metric Performance-Parameter Trade-off (PPT).

**Relation To Prior Work:**

The work discussed the existing PEFT benchmark and compared the proposed benchmark and the existing one.

**Summary And Contributions:**

This paper introduces a unified benchmark designed for evaluating parameter-efficient fine-tuning (PEFT) techniques across various vision tasks, including classification and video recognition, to facilitate fairer comparisons. The benchmark encompasses 30 diverse datasets and 25 state-of-the-art PEFT algorithms, offering a significantly more comprehensive evaluation than existing benchmarks. Additionally, the project provides well-organized open-source code and checkpoints for the trained models, reducing the effort required to utilize this benchmark.

---

> ### Author Rebuttal · Authors · 2024-08-17
>
> Dear Reviewer etT1,
>
> Thank you for taking the time to review our paper. Your insightful reviews are greatly appreciated. We have carefully considered them and provided detailed responses below.
>
> **Q1:** "Explain the insight of the proposed Performance-Parameter Trade-off (PPT) metric."
>
> **A1:** In our paper, we introduced the PPT metric in **Section 6.1**, providing a clear definition and initial explanation. We expanded on this further in **Appendix E**. However, we understand there may still be some confusion, and we would like to clarify the following points:
>
> **Motivation:** In PETL field, comparisons of proposed methods are typically based on the number of trainable parameters and performance on downstream tasks. However, these two criteria are usually evaluated independently, lacking a unified metric. To address this gap, we introduced the PPT metric. PPT provides a comprehensive and unified assessment of the strengths and weaknesses of various PETL methods, offering researchers a practical and efficient evaluative tool.
>
> **Innovation in PPT Design:** The innovation of PPT design has been described in detail in **Appendix E**. Here we provide further explanation. The main purpose behind the PPT design is to show how each increase in trainable parameters improves the performance of downstream tasks. This can be described simply as:
>
> &emsp;$PPT=\frac{performance}{parameters}$
>
> However, special attention is needed for the parameters because it ranges from 0M to 10M ($1M = 10^6$). We use a constant C ($10^7$) to scale it to the range of [0, 1]. According to the study [1], trade-off evaluation metrics typically require adjustments using logarithmic (log) and exponential (exp) functions, which we also incorporate into our PPT metric design. To avoid logarithmic values dropping to negative infinity, we add a constant of 1. Thus, the final PPT metric can be expressed as:
>
> &emsp;$PPT=performance \times exp(-log_{10}(\frac{parameters}{C}+1))$.
>
> **Practical Application:** We applied the PPT metric to evaluate 25 PETL methods across 30 datasets in our study. It effectively addresses the challenges associated with evaluating performance in the field of PETL. We believe the PPT will become a pivotal evaluation metric in the domain of visual PETL in the foreseeable future.
>
> **Future Optimization:** Currently, most of visual PETL methods primarily focus on the number of parameters and downstream task performance while few studies considers the GPU resources and the FLOPs parameter that you highlighted. If there is future demand in the field of visual PETL, we plan to further optimize the PPT metric to support and advance the development of the PETL field.
>
> **Q2:** "It would be better to use FLOPs as a comparison metric."
>
> **A2:** The primary reason we did not include FLOPs in our evaluation is that it has been rarely considered within the visual PETL field. However, we recognize the value of your suggestion and agree that incorporating a broader range of metrics could significantly enhance our benchmark. In response, we have calculated the FLOPs on FGVC task, and the results are as follows:
>
> | Method  	| CUB-200-2011 | NABirds | Oxford Flowers | Stanford Dogs | Stanford Cars | Mean | Params.| GLOPs |PPT |
> |-|-|-|-|-|-|-|-|-|-|
> | Full fine-tuning   | 87.3         | 82.7    | 98.8           | 89.4          | 84.5          | 88.5    | 85.8M  |   17.58 G     | -   |
> | Linear probing    | 85.3         | 75.9    | 97.9           | 86.2          | 51.3          | 79.3    | 0 M    |   17.58 G     | 0.79|
> | Adapter         | 87.1         | 84.3    | 98.5           | 89.8          | 68.6          | 85.7    | 0.41M   |  17.61 G       | 0.84|
> | AdaptFormer     | 88.4         | 84.7    | 99.2           | 88.2          | 81.9          | 88.5    | 0.46M   |  17.61 G       | 0.87|
> | Prefix Tuning     | 87.5         | 82.0    | 98.0           | 74.2          | 90.2          | 86.4    | 0.36M   |  17.90 G      | 0.85|
> | BitFit           | 87.7         | 85.2    | 99.2           | 86.5          | 81.5          | 88.0    | 0.10M    |   17.58 G     | 0.88|
> | VPT-Shallow     | 86.7         | 78.8    | 98.4           | 90.7          | 68.7          | 84.7    | 0.25M    | 18.30 G       | 0.84|
> | VPT-Deep       | 88.5         | 84.2    | 99.0           | 90.2          | 83.6          | 89.1    | 0.85M    | 18.30 G     | 0.86|
> | SSF            | 89.5         | 85.7    | 99.6           | 89.6          | 89.2          | 90.7    | 0.39M    |   17.58 G     | 0.89|
> | LoRA          | 85.6         | 79.8    | 98.9           | 87.6          | 72.0          | 84.7    | 0.77M    |   17.58 G     | 0.82|
> | GPS           | 89.9         | 86.7    | 99.7           | 92.2          | 90.4          | 91.7    | 0.66M    |   17.58 G     | 0.90|
> | HST           | 89.2         | 85.8    | 99.6           | 89.5          | 88.2          | 90.4    | 0.78M    |   19.02 G     | 0.88|
> | LAST          | 88.5         | 84.4    | 99.7           | 86.0          | 88.9          | 89.5    | 0.66M    |   19.47 G  | 0.87|
> | SNF           | 90.2         | 87.4    | 99.7           | 89.5          | 86.9          | 90.7    | 0.25M    |  17.68 G     | 0.90|
>
> From the results, we observe that the FLOPs do not increase for many methods such as BitFit, SSF, LoRA, and GPS because these methods do not change the original structure of the pre-trained model. Conversely, the FLOPs for Adapter, AdaptFormer, VPT, SNF, and Prefix-Tuning slightly increase due to the addition of small learnable modules or tokens. Additionally, the FLOPs for HST and LAST significantly increase, as these methods introduce additional parallel sub-networks to the original model. We will continue evaluating FLOPs for other tasks and include these results in the revised version of our paper.
>
> Thank you once again for your insightful feedback.
>
> Best wishes,
>
> All authors of Submission 88.
>
> [1] Liu et al. ELEVATER: A Benchmark and Toolkit for Evaluating Language-Augmented Visual Models. NeurIPS 2022.

---

> > ### Author Rebuttal · Authors · 2024-08-22
> >
> > Dear Reviewer #etT1,
> >
> > We are sorry to bother you. This comment originates from our worries.
> >
> > Given the limited time remaining for the rebuttal, we are eager to know if we have addressed your concerns.
> >
> > Thanks again for your reviewing, we are very happy to see your feedback, and address your further concerns.
> >
> > Best wishes,
> >
> > All authors of Submission 88.

---

> > > ### Comment · Reviewer_etT1 · 2024-08-22
> > > **Respone to rebuttal**
> > >
> > > Dear authors, thank you for addressing all my concerns. I will maintain the score.

---

### Official Review · Reviewer_sENS · 2024-07-23
**Review for 88**

**Rating:** 5
**Confidence:** 5
**Correctness:** Yes.
**Clarity:** Yes.

**Review:**

While this paper offers a comprehensive benchmark for PETL, it lacks critical documentation on how to run the provided code. Additionally, the training logs and checkpoints, which the authors claim to have included, are not available.

**Strengths:**

See Review.

**Additional Feedback:**

See Review.

**Documentation:**

No documentation is provided as to how to run the code.

**Ethics:**

No.

**Limitations:**

The authors did not discuss any limitations, although they claimed so in the checklist.

**Opportunities For Improvement:**

See Review.

**Relation To Prior Work:**

Yes.

**Summary And Contributions:**

The authors proposed the V-PETL Bench, a benchmark for Parameter-Efficient Transfer Learning, implementing 25 algorithms with 30 datasets.

---

> ### Author Rebuttal · Authors · 2024-08-17
>
> Dear Reviewers #sENS,
>
> We thank you for your valuable comments. Here we address your concerns as follows.
>
> **Q1:** “While this paper offers a comprehensive benchmark for PETL, it lacks critical documentation on how to run the provided code. Additionally, the training logs and checkpoints, which the authors claim to have included, are not available.”
>
> **A1:** We will address your concerns as below:
>
> **Documentation:** Thank you for highlighting the importance of clear documentation. We agree that thorough documentation is essential for others to effectively use our benchmark. **We have added a comprehensive documentation in our GitHub repository.** The link to the Github repository is on the first page of our paper. This document includes an introduction, the running environment, download links to the dataset and pre-trained models, as well as links to the checkpoints and logs we have generated. Additionally, it provides the results of each method and detailed examples on how to add new methods to our benchmark. We hope that this detailed documentation will make it easier for others to use and extend our benchmark for their own research.
>
> **Checkpoints and Logs:** **We have uploaded the checkpoints and available logs to HuggingFace, and the link is provided in our GitHub repository.** These resources will enable others to easily reproduce our results. It takes a long time to organize these resources. Specifically, for 25 PETL algorithms, 30 datasets, and 5 pre-tained models, we need to provide 25 * 30 * 5 = 3750 checkpoints, and a lot of space is needed to store them.
>
> **Q2:** The authors did not discuss any limitations, although they claimed so in the checklist.
>
> **A2:** We have disscussed the limitations of V-PETL Bench in **Appendix H (Future Work)**. Specifically, we discuss two aspects:
>
> - V-PETL Bench currently focuses on traditional computer vision (CV) tasks. However, the PETL community should also consider other popular CV tasks such as text-to-image generation and point cloud analysis.
>
> - V-PETL currently supports only a limited selection of pre-trained models. It is crucial to expand our repository to include self-supervised and multimodal pre-trained models.
> For more details, please refer to Appendix H.
>
> Thank you once again for your insightful feedback.
>
> Best wishes,
>
> All authors of Submission 88.

---

> > ### Author Rebuttal · Authors · 2024-08-22
> >
> > Dear Reviewer #sENS,
> >
> > We are sorry to bother you. We hope our edits addressed your comments.
> >
> > Please let us know if there are any additional clarifications or edits you would advise.
> >
> > Thank you once again for your feedback.
> >
> > Best wishes,
> >
> > All authors of Submission 88.

---

### Official Review · Reviewer_y7Gu · 2024-07-24

**Rating:** 7
**Confidence:** 4
**Correctness:** Yes
**Clarity:** Yes

**Review:**

V-PETL contains extensive evaluation of different PETL algorithms on multiple datasets. This is so far the largest benchmark for visual parameter-efficient transfer learning algorithms and will benefit the community. The author also proposed a new metric called Performance-Parameter Trade-off (PPT) for comparing different PETL algorithms and visualized the learned feature maps.

**Strengths:**

- Comprehensive evaluation of a large collection of PETL methods on multiple vision tasks
- Unified comparison metric based on Performance-Parameter Trade-off (PPT)

**Additional Feedback:**

N/A

**Documentation:**

Yes

**Ethics:**

No.

**Limitations:**

Yes

**Opportunities For Improvement:**

The paper is a solid contribution and I haven't seen major angles for improvement.

**Relation To Prior Work:**

Yes

**Summary And Contributions:**

The paper proposed V-PETL Bench. V-PETL bench compared 25 parameter-efficient transfer learning (PETL) algorithms on 30 datasets that cover image recognition, video action recognition and dense prediction. Apart from reporting performance numbers, the author also analyzed the computational cost of different PETL algorithms, and visualized the feature distribution after trained by different models.

---

> ### Author Rebuttal · Authors · 2024-08-12
>
> Dear Reviewer #y7Gu:
>
> We sincerely appreciate the time and effort you have invested in reviewing our paper. We are particularly grateful for your recognition of the significant contribution our work makes, especially regarding the **comprehensive evaluation** and the **unified comparison metric** we introduced.
>
> Should there be any further questions or concerns in the future, we would be more than pleased to address them.
>
> Thank you once again for your insightful feedback.
>
> Best wishes,
>
> All authors of Paper 88.

---

> > ### Author Rebuttal · Authors · 2024-08-22
> >
> > Dear Reviewer #y7Gu:
> >
> > Thank you again for your recognition of our work. We are committed to further enhancing and maintaining the V-PETL Benchmark to contribute to the advancement of the PETL field.
> >
> > As the time for rebuttal is drawing to a close,  please let us know if you have any additional questions or concerns.
> >
> > We are ready to provide further clarifications and eagerly await your final score.
> >
> > Best wishes,
> >
> > All authors of Paper 88.

---

### Official Review · Reviewer_Pejx · 2024-07-31
**V-PETL Bench: A Unified Visual Parameter-Efficient Transfer Learning Benchmark**

**Rating:** 7
**Confidence:** 4
**Correctness:** Yes.
**Clarity:** Yes.

**Review:**

Pros:
1. The paper presents a well-structured and comprehensive benchmark for visual parameter-efficient transfer learning (PETL). The methodology is clearly outlined, and the evaluation of 25 algorithms across 30 datasets demonstrates thoroughness and attention to detail.
2. The paper is well structured and well written.
3. This benchmark covers a wide range of datasets and tasks, providing a robust platform for evaluation.
4. The modular and extensible codebase promotes reproducibility and encourages further research.

Cons:
The evaluation requires significant computational resources (310 GPU days), which may limit accessibility for some researchers. It may cause difficulty for researchers to reproduce or extend.

**Strengths:**

Please check the pros.

**Additional Feedback:**

NA

**Documentation:**

Yes.

**Ethics:**

No.

**Limitations:**

Yes.

**Opportunities For Improvement:**

Please check the cons.

**Relation To Prior Work:**

Yes.

**Summary And Contributions:**

The paper introduces V-PETL Bench, the first comprehensive benchmark for visual parameter-efficient transfer learning (PETL). It evaluates 25 dominant PETL algorithms across 30 computer vision datasets, providing a structured framework for comparing their performance. The key contributions of the paper include:

1. Benchmarking Framework: V-PETL Bench serves as a standardized platform for assessing the effectiveness of various PETL algorithms in visual tasks, facilitating easier comparisons and evaluations.

2. Performance Metrics: The authors propose the Parameter Performance Trade-off (PPT) metric, which allows for a nuanced comparison of algorithms based on both the number of parameters and their performance.

3. Insights and Analyses: The paper presents insightful analyses of the benchmark results, highlighting the strengths and weaknesses of different PETL algorithms, particularly in relation to full fine-tuning methods.

4. Modular Codebase: The codebase is organized into four layers (Core, Algorithm, Extension, and API), making it user-friendly and extensible for researchers interested in applying PETL algorithms to new applications.

Overall, the paper establishes V-PETL Bench as a significant resource for advancing research in parameter-efficient transfer learning within the visual domain.

---

> ### Author Rebuttal · Authors · 2024-08-16
>
> Dear Reviewers #Pejx,
>
> We sincerely appreciate your valuable feedback and recognition of our work's contribution. We have carefully addressed each of your questions and detailed our responses below.
>
> **Q1:** "The evaluation requires significant computational resources (310 GPU days), which may limit accessibility for some researchers. It may cause difficulty for researchers to reproduce or extend. "
>
> **A1:** We will address your concerns as below:
>
> **Computational Resources Analysis:** In this work, we have developed a comprehensive PEFT benchmark, which covers the mainstream datasets, the pre-trained models, and the most representative methods in PEFT filed. Consequently, it is typical to require 310 GPU days for a study of this scale. Additionally, other domain-specific benchmarks usually require similar GPU computational resources. For example, the semi-supervised benchmark TorchSSL [1], takes approximately 335 GPU days.
>
> **Accessibility:** While our benchmark is broad, researchers often focus on specific tasks rather than the entire benchmark. For instance, most PETL studies [2-4] concentrate on image classification task on the VTAB dataset, which only require around 1 GPU day.
>
> **Reproducibility:** As you mentioned, training models from scratch can be both time-consuming and resource-intensive. We hope to provide convenience to researchers instead of making it more time-consuming. Thus, to reproduce our results, researchers can just load our checkpoints and the download link is available in our Github repository. This will allow users to reproduce our results without extensive retraining, thereby reducing the computational burden significantly.
>
> **Extendability:** The V-PETL Benchmark was designed with future expansion in mind. It supports the addition of new PETL methods, pre-trained models, or datasets through a modular interface that aligns with the existing components. We have also provided comprehensive documentation and examples to assist users in extending the benchmark with new elements, ensuring a user-friendly integration process.
>
> Thank you once again for your insightful feedback.
>
> Best wishes,
>
> All authors of Submission 88
>
> [1] Zhang et al. Flexmatch: Boosting semi-supervised learning with curriculum pseudo labeling. NeurIPS 2021.
>
> [2] Lian et al. Scaling & Shifting Your Features: A New Baseline for Efficient Model Tuning. NeurIPS 2022.
>
> [3] Jie et al. FacT: Factor-Tuning for Lightweight Adaptation on Vision Transformer. AAAI 2023.
>
> [4] Zhao et al. SCT: A Simple Baseline for Parameter-Efficient Fine-Tuning via Salient Channels. IJCV 2024.

---

> > ### Author Rebuttal · Authors · 2024-08-22
> >
> > Dear reviewer,
> >
> > Thank you again for taking the time to review our work.
> >
> > Please let us know if you have any concerns left after our response. We would be happy to discuss any further questions and comments you may have.
> >
> > Thank you once again for your feedback.
> >
> > Best wishes,
> >
> > All authors of Submission 88

---

### Author Rebuttal · Authors · 2024-08-17

Dear Reviewers and Area Chairs,

We thank all reviewers for their constructive thoughtful feedbacks. The reviewers agree that:

**Contribution to PETL field:**

- **Reviewer #Pejx:** "The paper establishes V-PETL Bench as a significant resource for **advancing research** in PETL within the visual domain."

- **Reviewer #y7Gu:** "This is so far the largest benchmark for visual PETL algorithms and will **benefit the community**."

- **Reviewer #etT1:** "I foresee this benchmark playing **a pivotal role** in the field of PEFT."

**Novel and Comprehensive Benchmark:**

- **Reviewer #Pejx:** "The paper presents a well-structured and **comprehensive benchmark** for visual PETL and this benchmark covers a wide range of datasets and tasks, providing a robust platform for evaluation."

- **Reviewer #y7Gu:** "**Comprehensive evaluation** of a large collection of PETL methods on multiple vision tasks."

- **Reviewer #sENS:** "This paper offers a **comprehensive benchmark** for PETL".

- **Reviewer #etT1:** "The V-PETL offers a significantly **more comprehensive evaluation** than existing benchmarks.".

**Unified and Effective Evaluation Metric:**

- **Reviewer #Pejx:** "The authors propose the **Parameter Performance Trade-off (PPT) metric**, which allows for a nuanced comparison..."

- **Reviewer #y7Gu:** "**Unified comparison metric** based on Performance-Parameter Trade-off (PPT)"

**Well-Written and Organized:**

- **Reviewer #Pejx:** "The paper is well structured and well written."

- **Reviewer #y7Gu:** "The paper is a solid contribution and I haven't seen major angles for improvement."

- **Reviewer #etT1:** "The paper is well-organized and easy to follow."

We are deeply encouraged by these positive comments and appreciate all reviewers for their recognition of our work.

Additionally, we have addressed each reviewer's questions individually. Specifically, we have provided detailed descriptions of **computational resources and extensibility**, **documentation and checkpoints (available in our Github repository)**, and **the insights into the PPT metric**.

Regarding the experiments, we have included statistics on the **FLOPs metric**, which further enhances our V-PETL benchmark.

Thank you again for your time and effort in reviewing our paper!

Best wishes,

All authors of Submission 88.

---

### Decision · Program_Chairs · 2024-09-26

**Decision:**

Accept (Poster)

**Comment:**

This paper received mostly positive reviews from four reviewers (8,7,7,5). The one negative review was quite short, with little information beyond criticism about missing documentation logs and checkpoints, which has since been appropriately addressed by the authors with updates to their GitHub repository. Beyond that, the other reviewers are appreciative of the comprehensive, large-scale effort the authors delivered in producing this benchmark and code base, as well as their analysis of existing PETL methods. I recommend acceptance.